# Revealing and reshaping attractor dynamics in large networks of cortical neurons

**Chen Beer** [1,2], **Omri Barak** [2,3]*

**1** Andrew and Erna Viterbi Faculty of Electrical and Computer Engineering, Technion - Israel Institute of Technology, Haifa, Israel, **2** Network Biology Research Laboratories, Technion - Israel Institute of Technology, Haifa, Israel, **3** Rappaport Faculty of Medicine, Technion - Israel Institute of Technology, Haifa, Israel

* omri.barak@gmail.com

**Data Availability Statement:** All code and data can be accessed via https://github.com/Chen-Beer/Beer-Barak-2023.

**Funding:** OB is supported by the Israeli Science Foundation (grant 1442/21) and an HFSP research

## Abstract

Attractors play a key role in a wide range of processes including learning and memory. Due to recent innovations in recording methods, there is increasing evidence for the existence of attractor dynamics in the brain. Yet, our understanding of how these attractors emerge or disappear in a biological system is lacking.

By following the spontaneous network bursts of cultured cortical networks, we are able to define a vocabulary of spatiotemporal patterns and show that they function as discrete attractors in the network dynamics. We show that electrically stimulating specific attractors eliminates them from the spontaneous vocabulary, while they are still robustly evoked by the electrical stimulation. This seemingly paradoxical finding can be explained by a Hebbian-like strengthening of specific pathways into the attractors, at the expense of weakening non-evoked pathways into the same attractors. We verify this hypothesis and provide a mechanistic explanation for the underlying changes supporting this effect.

## Author summary

There are many hints that could evoke the same memory. There are many chains of evidence that could lead to the same decision. The mathematical object describing such dynamics is called an attractor, and is believed to be the neural basis for many cognitive phenomena. In this study, we aimed to deepen our understanding of the existence and plasticity of attractors in the dynamics of a biological neural network. We explored the spontaneous activity of cultured neural networks and identified a set of patterns that function as discrete attractors in the network dynamics. To understand how these attractors evolve, we stimulated the network to repeatedly visit some of them. Surprisingly, we observed that the stimulated patterns became less common in the spontaneous activity, while still being reliably evoked by the stimulation. This paradoxical finding was explained by the strengthening of specific pathways leading to these attractors, alongside the weakening of other pathways. These findings provide valuable insights into the mechanisms underlying attractor plasticity in biological neural networks.

grant (RGP0017/2021). The funders had no role in study design, data collection and analysis, decision to publish, or preparation of the manuscript.

**Competing interests:** The authors have declared that they have no competing interests.

## Introduction

Attractors are important elements in many cognitive processes such as memory formation and decision-making. These attractors are considered to arise from the dynamics of neuronal networks in the brain, which allow for the emergence of stable states that can persist over time. For instance, head-direction circuits need to integrate body motion over time, consistent with continuous attractor dynamics [1, 2]. Working memory of discrete [3] or continuous [4] information was hypothesized to be supported by attractors [5]. Decision-making can be interpreted as convergence to a discrete set of attractors [6], and many other examples exist [7]. Nevertheless, despite their key role in brain function, the mechanisms underlying the generation of such attractors and their evolution over time remain largely unknown.

To address this challenge, we focus on the relationship between spontaneous and evoked activity [8]. Attractors, as the name implies, attract neural activity from nearby starting points into a common trajectory. This set of initial conditions is known as a basin of attraction. If attractor dynamics are relevant for behavior, one would expect external stimuli to lead neural activity into one of these basins. Similarly, it is reasonable to expect spontaneous activity to occasionally land into one of the basins, and hence result in the activation of attractors. In line with these expectations, there have been reports of spontaneous reactivations that are similar to evoked activity [9–11].

We studied this question in a more controlled setting—using in-vitro cultured cortical neurons. These networks can sustain both spontaneous [12] and evoked [13] activity, and allow continuous monitoring over many hours. Furthermore, it was shown that structured stimulation can lead to learning in such networks [14].

In this paper, we show that the spontaneous activity of in-vitro cortical networks contains a vocabulary of spatiotemporal patterns that act as discrete transient attractors. Discreteness is manifested by the finite number of such patterns that repeat over time. We show that nearby initial conditions lead to the same pattern, consistent with basins of attraction. These attractors are transient, as these network bursts are of limited duration, and the network relaxes to a quiescent state following each burst. Furthermore, we demonstrate that specific localized stimulation can generate robust evoked responses from this vocabulary of attractors. We also show that prolonged stimulation of these specific attractors leads to their elimination from the spontaneous vocabulary, while still being robustly evoked by the stimulation.

This work provides the first direct evidence for the plasticity of multiple attractors in a biological neural network. In addition, the plasticity principles described in the paper improve our understanding of how attractors in a biological system evolve. This study sheds light on the mechanisms underlying attractor dynamics in the brain and offers a new perspective on how they can be manipulated.

## Results

To study attractor dynamics, we use extracellular recordings of mature networks of cultured cortical neurons (18–21 DIV, see Methods). Electrical activity is recorded from a multi-electrode array (MEA) of 120 electrodes on which the neurons are plated (Fig 1A). Throughout the following sections, we will demonstrate the results using one example experiment, and show statistics across all experiments. Further details regarding all experiments are in the methods section.

### Spontaneous vocabulary as attractor dynamics

One of the main characteristics of the activity of cultured neuronal networks is the presence of spontaneous synchronized bursts (network bursts) [15–17], in which a large fraction of the

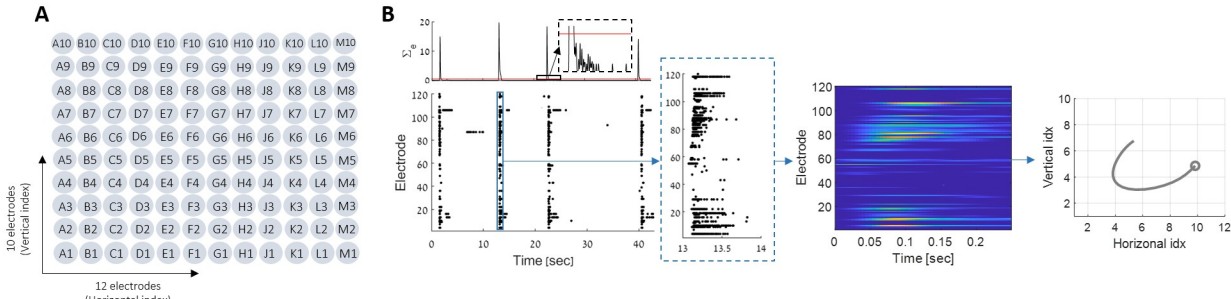

**Fig 1. Network burst extraction.** **(A)** Electrode layout in the MEA. The 120 electrodes are arranged in a 12x10 array, spaced 1mm vertically and 1.5mm horizontally. See Methods for more details. **(B)** The activity of all electrodes (bottom-left) is summed into a one-dimensional time series (top-left). A threshold is used to define a burst (red line). The analysis is done after smoothing and binning the data, to get a continuous time series for each electrode (heatmap, see Methods). For visualization only, we use the center of mass (COM) representation in the MEA physical space (10x12 electrodes)—the axes represent the electrodes' indices (rightmost plot, circle denotes initial state. See Methods).

neurons fire almost simultaneously within a few hundred milliseconds (Fig 1B). We follow the spontaneous activity of matured cultured cortical neuronal networks, focusing on these bursting events. To do so, we record multi-unit activity from an array of 120 electrodes (Fig 1B, bottom). We define these network bursts based on the summed activity across all electrodes (Fig 1B, top), beginning with a threshold-crossing, and ending when the same threshold is crossed again. A burst can be described as a spatiotemporal pattern (Fig 1B, heat-map), or as a trajectory in the 120-dimensional space of the neural activity. For visualization purposes, we take advantage of the physical ordering of the electrodes, and project these trajectories to a natural two-dimensional space (Fig 1A)—the physical location of the activity's center of mass (Fig 1B, rightmost plot). Cultured networks emitted such bursts at a rate of 8 ± 4 per minute (see Tables 1 and 2). Although a large fraction of the electrodes are recruited by every burst, they are not all alike. We noticed that each network has its own repertoire of such spatiotemporal patterns —a finite set of network bursts that repeat many times spontaneously (Fig 2C).

Looking more closely at these bursts reveals attractor-like dynamics: Similar initial conditions lead to similar bursts. To see this, we define the initial condition of a burst as the spatial activity at the moment of threshold crossing—a vector in 120 dimensions. For a pair of bursts, we can measure the similarity of initial conditions by correlating these vectors. We can also

**Table 1. Stimulation experiments.** A detailed description of every column can be found in the methods.

| MEA # | Prep date | Age (DIV) | Before stimulation | | | Stimulated existence (out of 3) | After stimulation | | |
|---|---|---|---|---|---|---|---|---|---|
| | | | Bursts/hour | # of clusters | % Explained | | Bursts/hour | # of clusters | % Explained |
| 26550 | 1.11 | 19 | 260 | 9 | 85 | 3 | 530 | 8 | 83 |
| 26549 | 8.11 | 20 | 292 | 14 | 98 | 1 | 651 | 11 | 97 |
| 38428 | 17.11 | 18 | 274 | 18 | 95 | 1 | 686 | 20 | 85 |
| 38427 | 1.11 | 20 | 694 | 13 | 87 | 2 | 739 | 14 | 92 |
| 26532 | 2.3 | 20 | 497 | 16 | 89 | 1 | 463 | 16 | 87 |
| 26550 | 3.5 | 19 | 617 | 17 | 93 | 1 | 563 | 18 | 89 |
| 38426 | 2.11 | 19 | 691 | 14 | 90 | 1 | 688 | 19 | 92 |
| 26549 | 11.11 | 19 | 679 | 7 | 66 | 1 | 605 | 8 | 98 |
| 26550 | 15.11 | 20 | 626 | 18 | 97 | 2 | 647 | 10 | 95 |
| N/A | 8.11 | 21 | 548 | 12 | 89 | 2 | 607 | 7 | 88 |
| 38428 | 20.2 | 21 | 301 | 7 | 96 | 3 | 392 | 8 | 94 |

**Table 2. Control experiments.** A detailed description of every column can be found in the methods.

| MEA # | Prep date | Age (DIV) | Before stimulation | | | Stimulated existence (out of 3) | After stimulation | | |
|---|---|---|---|---|---|---|---|---|---|
| | | | Bursts /hour | # of clusters | % Explained | | Bursts/hour | # of clusters | % Explained |
| 26550 | 24.1 | 21 | 553 | 11 | 81 | 3 | 664 | 8 | 71 |
| 39740 | 24.4 | 18 | 609 | 17 | 88 | 1 | 659 | 10 | 79 |
| 38427 | 24.4 | 21 | 632 | 18 | 97 | 3 | 600 | 16 | 85 |
| 26536 | 7.2 | 20 | 234 | 15 | 94 | 3 | 453 | 17 | 83 |
| 38427 | 7.2 | 21 | 566 | 9 | 84 | 2 | 631 | 17 | 88 |

measure the similarity between entire bursts, by flattening them and then obtaining a correlation coefficient (2D correlation, see Methods). Repeating this for all burst pairs allows us to look at the joint distribution of these two measures (Fig 2A). We find that this distribution is bimodal, allowing us to define a threshold on the similarity of initial conditions ($\theta$, horizontal dashed line) that will lead to similar overall bursts (vertical dashed line). Note that $\theta$ is a network-specific threshold, depending on the distribution. Conversely, we see that most pairs of bursts are much less similar—indicating the presence of more than one attractor. Further support to the attractor dynamics is given by the convergence of similar bursts over time. If we consider all pairs of highly correlated bursts (2D correlation above $\theta$) and compute their instantaneous correlation, we see that the variability between them decreases over time (Fig 2B). We conclude that bursts can be described as distinct attractors, each with its own basin of attraction.

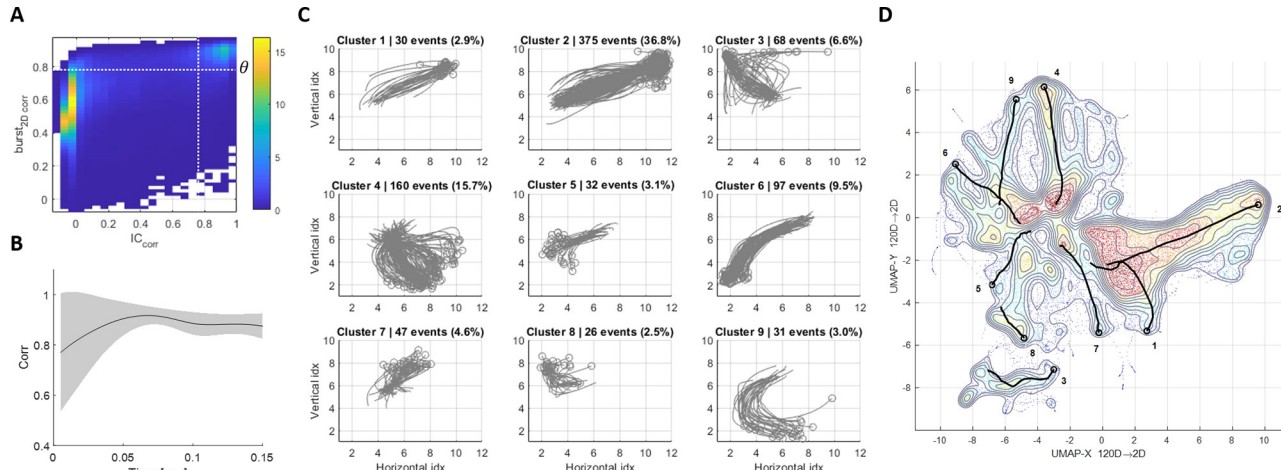

**Fig 2. Attractor dynamics. (A)** Similar initial states of bursts lead to similar bursts (top-right). 2-D histogram of two correlation measures across all pairs of bursts—spatial correlation between their initial states and the 2D correlation between the full bursts. Initial states are defined as the activity at the first 5msec of a burst, after the threshold crossing. White dashed lines denote the thresholds used for clustering (see Methods). The horizontal threshold, $\theta$, is used as a similarity threshold between bursts. **(B)** Dynamics of convergence. For every pair of similar bursts (2D correlation larger than $\theta$), we measure the spatial correlation at every point in time. The black line denotes the mean and the grey shade represents the standard deviation (not SEM). Note the diminishing variability with time, indicating convergence from variable initial states. **(C)** Dynamics-based clustering. In this example, the vocabulary contains 9 main clusters, explaining about 85% of the spontaneous bursts. Each subtitle contains the number of bursts in each cluster and the percentage out of all spontaneous bursts. Each center of mass trajectory (grey) is a single burst, the circles denote the initial state of each burst. Axes represent the electrodes' indices. **(D)** UMAP embedding representing all the 1017 spontaneous bursts recorded during 4 hours of activity. Contours and background colors denote the density of neural states (blue—low density, red—high density). The black solid lines represent the median trajectory of each of the 9 clusters in the spontaneous vocabulary (the cluster numbers are noted next to the trajectories' onset). The observed tendency of bursts to move toward the center is primarily due to the fact that the center of the UMAP space corresponds to the origin (0,0), and the activity of the bursts diminishes as they progress towards their end.

To enumerate these attractors, we cluster all bursts into distinct groups. In the case of the example network, 1017 bursts were divided into 9 clusters, with 151 bursts left unclassified. Fig 2C shows these clusters in the MEA physical space. Clustering was based on a similarity graph between all bursts, where similarity was defined by three different measures (see Methods). We then use spectral clustering ([18], see Methods) to define a vocabulary of attractors for each network.

The center-of-mass (COM) trajectories are two-dimensional projections of a 120-D space, and do not capture the full phase space of neural activity. To provide another view of neural activity, we use a non-linear dimensionality reduction method (UMAP [19]). We consider all high-D neural states of all 1017 bursts, disregarding their temporal structure. Projecting these states into the first two dimensions of UMAP yields the contour plots seen in Fig 2D. Overlaid on this plot are the medians of each cluster, showing that they are mostly separated in neural activity, despite having large overlaps in the COM projection. Note that the UMAP projection was not used to define clusters, and is thus an independent view of the attractor phenomenon.

Attractors in dynamical systems describe areas of phase space to which activity converges, and does not leave. In contrast, the bursts we describe are transient events. Nevertheless, we can think of them as attractors of the dynamical system until the peak of the burst. Once the burst is established, the dynamics change (probably due to adaptation), and the attractor destabilizes. Alternatively, one can consider a single global attractor—the quiescent state. In this interpretation, the basin of attraction of this single attractor is highly structured. Each of the bursts is a specific pathway within this basin, that is separated from the others. This separation of timescales is common in the analysis of dynamical systems. We are interested in the fast attraction phase of these dynamics, and not in the slow relaxation from them, and will thus refer to these bursts as discrete transient attractors.

## Evoked responses

We showed the existence of attractors using spontaneous activity. The motivation to study attractors, however, stems from evoked activity. Attractors have been suggested to support the memory of stimuli [20], to maintain a decision until it is carried out [21, 22], or to support other computations related to evoked activity. Previous studies in-vivo showed conflicting accounts on the relationship between spontaneous and evoked activity [10, 11]. We explored this question in our controlled settings. Namely, we asked whether spontaneous and evoked activity reside in the same dynamical landscape.

First, we searched for stimulation sites that generate a robust response. To this end, we divided the MEA into 20 stimulation sites: sets of 6 adjacent electrodes (organized in a 3x2 configuration, with no overlap) that span the entire 2D MEA space (Fig 3A). We then tested their robustness by injecting a simultaneous voltage pulse to all 6 electrodes. We repeated this for the 20 sites for 30 cycles, with 10 seconds between each stimulation (see Methods).

Some of the stimulation sites generated a robust response (site 17, Fig 3B, bottom), while other sites did not (site 10, Fig 3B, top). We quantified the robustness of a response to each site by calculating the pairwise 2D correlation between all its 30 responses (Fig 3C). Focusing on the robust responses, we can ask whether they are part of the spontaneous vocabulary of the network or whether they represent an entirely different dynamics. For the robust response of site 17 shown in Fig 3B, we see that the correlation within different repetitions of the evoked response is as strong as the correlation between 17's evoked responses and the spontaneous bursts belonging to one of the spontaneous attractors (cluster 5, Fig 3D). The similarity between spontaneous and evoked in this case can also be appreciated via the center of mass trajectories (Fig 3F), which also emphasize the different initial conditions. Note that the

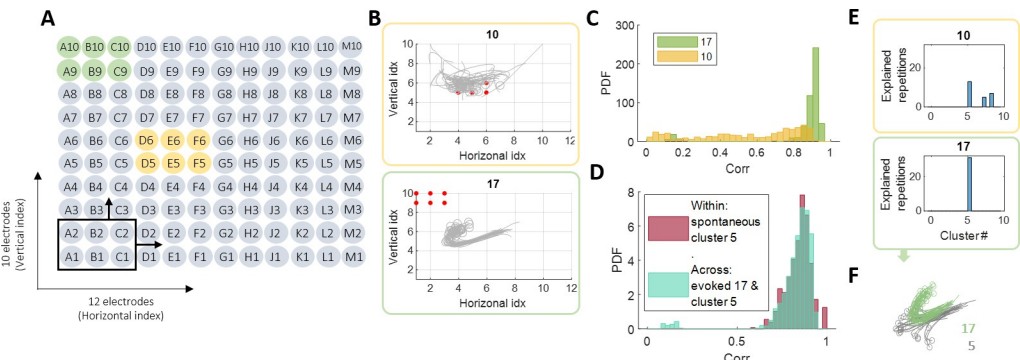

**Fig 3. Evoked responses. (A)** Electrode arrangement in the MEA (120 in total). Each stimulation site consists of 6 adjacent electrodes (black frame). There are 20 such stimulation sites with no overlapping electrodes. The set of 6 electrodes colored in green represents stimulation site number 17, and the set in yellow refers to site number 10. Probe stimulation: We stimulated all sites, one after the other, 30 times each. The time between stimuli is 10 seconds. This lasts 1.75 hours in total. **(B)** Visualization of the evoked responses to sites 10 and 17: center of mass representation for all of the 30 responses to each of these sites. The red dots denote the 6 stimulating electrodes in each case. **(C)** Robustness: probability density function (PDF) of the pair-wise correlation values within each of the 2 sites. It is clear that the network response to stimulation at site 17 is much more robust and coherent compared to 10. **(D)** Existence: Comparing the evoked responses to the spontaneous vocabulary. Here we show the PDF of the pair-wise correlation values within cluster 5 in the spontaneous activity and the PDF of the pair-wise correlation values between the evoked responses to site 17 and the spontaneous bursts in cluster 5. They overlap almost completely, meaning that the evoked responses to 17 are indeed part of the spontaneous vocabulary of the network. **(E)** Existence in terms of spontaneous vocabulary: Which spontaneous clusters explain the 30 evoked responses for each of the 2 sites? In the case of site 10—there is no specific cluster, also—a large part of the responses is not explained by any of the clusters. In the case of site 17, cluster number 5 explains all of the evoked responses. **(F)** Center of mass trajectories of the evoked responses to 17 (green) together with the spontaneous bursts in cluster 5 (grey). This illustrates the convergence of the two classes to the same attractor, and emphasizes the different initial states they start from.

stimulating electrodes themselves are excluded from this analysis to avoid artifacts (see Methods). We thus see that cluster 5 *explains* the evoked responses to site 17, a term which we will define and use for all stimulation sites.

To systematically measure the existence of all evoked responses in the spontaneous clusters, we define a cluster as *explaining* a specific repetition of a specific stimulation site. This requires many of the spontaneous events of the cluster to be similar to that specific repetition (see Methods). Fig 3E shows which clusters explain the 30 responses of sites 10 and 17. Some of the evoked responses don't align with any specific cluster, resulting in histograms that may not add up to 30. If the stimulation site led the network to the basin of attraction of one of the spontaneous clusters, we expect a histogram similar to that of site 17 (Fig 3E, green frame). It could also be that the evoked response is at the border of a few basins, and thus more than one spontaneous cluster is needed to account for all 30 bursts.

Our results allow us to use the stimulation sites which generated a robust response as switches to control the dynamics of the network—we can now force the network to visit specific areas in the dynamical space. This raises the following questions: What will happen to the network's evoked response to this stimulation? What will happen to the spontaneous dynamics? Will the spontaneous vocabulary change? What will happen to the stimulated attractors in comparison to the non-stimulated ones?

## Strengthening and weakening specific pathways

In order to answer these questions, we use the following protocol (Fig 4A): We record the spontaneous activity of the network for four hours (during which the dynamics is stable), then

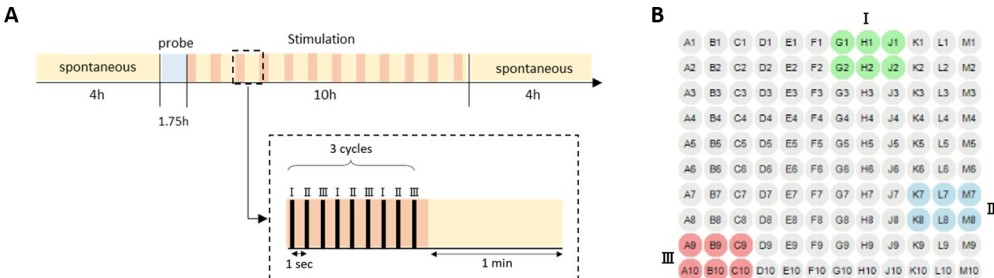

**Fig 4. Experiment protocol. (A)** Each experiment starts with a 4-hour recording of spontaneous activity. We then probe the system in 20 different stimulation sites, one after the other, and analyze the evoked responses to each of the 20 sites. We choose the 3 stimulation sites which generated the most robust and distinct evoked responses, and start a 10-hour stimulation period in which we alternate between stimulating these 3 sites (I, II, and III) and recording spontaneous activity (see inset). Following the 10-hour stimulation, we record the spontaneous activity for another 4 hours. Control experiments in which the 10-hour stimulation period had no stimulation but only spontaneous activity recordings, were also done (see Table 2). **(B)** The 3 selected stimulation sites, I, II, and III, in the experiment presented throughout this manuscript (site 3 in green, site 16 in blue, and site 17 in red).

we test the 20 stimulation sites as described above (Fig 3A). For each network, we select the three most robust stimulation sites using visual inspection of the center of mass trajectories, and correlation histograms (see Methods). These sites, denoted I, II, and III, are then stimulated for 10 hours (Fig 4). Finally, we record the spontaneous activity of the network again for an additional four hours.

The selection of the sites for prolonged stimulation was based on a short preliminary analysis conducted following the probe phase of each experiment (see Methods). The existence and robustness analysis (Fig 3B–3E) was done after the entire experiment was over. Therefore, before analyzing the changes in the spontaneous activity after stimulation, we have to make sure that the stimulation sites we selected do indeed lead to robust bursts that also exist in the spontaneous vocabulary of the network. To do so, we repeat the analysis done in Fig 3 for all networks and all stimulation sites. For each stimulation site we evaluate the robustness of its evoked responses (Fig 5, robustness) using the correlation among the 30 repetitions. We quantify the existence of this response in the spontaneous vocabulary by asking how concentrated histograms like those of Fig 3E are. Namely, how many spontaneous clusters are required to explain the majority of the 30 evoked responses of a particular stimulation site (Fig 5, existence). We see that robust responses (high values) are mostly associated with existing bursts (low values) from the spontaneous vocabulary of the network. Since this analysis was done after the entire experiment was over, not all selected sites (denoted in red circles) were indeed robust and similar to spontaneous bursts. Our analysis will focus on those that, post-hoc, were found to pass these criteria (green background).

We expected this protocol to target three (or less) attractors and strengthen them in a Hebbian manner. Namely, the evoked responses will be more robust, and the corresponding spontaneous patterns will be more present in the spontaneous activity. Surprisingly, we observed two opposite effects: the spontaneous activity linked to stimulation weakened, while the evoked responses did become more robust.

To quantify the changes in the spontaneous vocabulary, we asked what happened to the spontaneous bursts that were similar to the evoked ones, following the prolonged stimulation. For instance, we can correlate the spontaneous bursts in cluster 5 mentioned above (Fig 3E) to all 1017 spontaneous bursts that occurred before stimulation. The histogram in Fig 6A (blue) shows a large peak in high correlation values, consistent with the fact that this pattern is part of

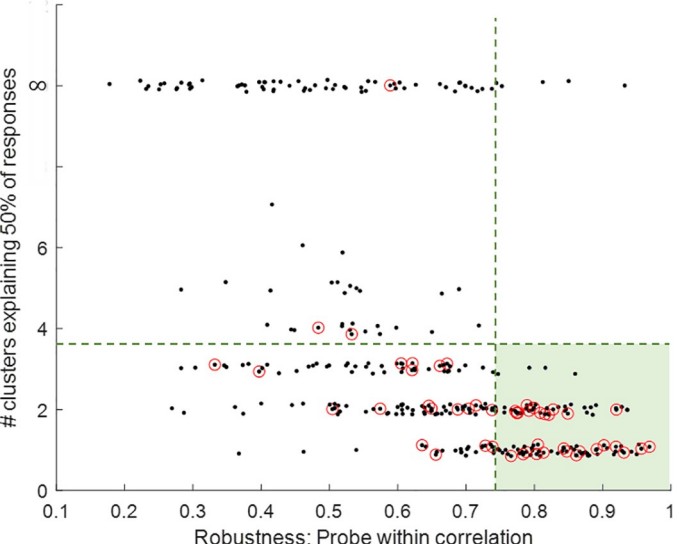

**Fig 5. Existence and robustness of evoked responses.** Each black dot represents a probe site in a given experiment (20 sites, 16 experiments, see Methods). The X-axis denotes the median pair-wise correlation values between the 30 responses. The Y-axis represents a measure of existence in the spontaneous vocabulary: the number of clusters required to explain 50% of the 30 responses. Small random noise is added to the number of clusters to aid in visualization. In general, the more robust the response is—fewer clusters are required to explain it. Red circles denote the stimulation sites selected for prolonged stimulation (3 sites in each experiment, see Fig 4). If at least 50% of the 30 responses exist in one or more of the spontaneous clusters, the corresponding number appears on the Y-axis. If not, the number is represented as infinity, which explains the jump in the graph after approximately 6 clusters. The evoked responses in the green area were defined as robust and existing in the spontaneous vocabulary, and therefore are used for further analysis.

the vocabulary. Repeating the same analysis, but this time comparing to the 2092 spontaneous bursts from the period after the stimulation, results in a very different distribution (Fig 6A, purple). We can see that the stimulated attractor almost disappeared from the spontaneous vocabulary.

We quantified the changes in the existence of patterns using the cumulative probability distribution, exemplified in Fig 6B for the two distributions mentioned above. Intuitively, we care about changes in high values of correlation—as these indicate spontaneous bursts that are similar to the pattern of interest. The exact definition of *high* is somewhat arbitrary, which is why we use a range of threshold values ($\alpha$, see Methods). Using this threshold, we can calculate the change in the existence of high-correlation patterns (Fig 6B).

Is this change due to our stimulation or simply a result of drift over time? We repeated this analysis for 11 networks with stimulated patterns (Table 1), and for 5 networks without stimulation (Table 2). Importantly, for these 5 control networks, we also chose 3 robust patterns but simply did not stimulate them. We see that the stimulated patterns tend to disappear from the spontaneous vocabulary after stimulation ($\Delta CDF$ is negative), while the mean effect in the control experiments is smaller (Fig 6C). Note that the difference between control and stimulation experiments is not statistically significant, but the trend in several analysis methods is in the same direction, with some of them reaching p-values of 0.05 (see S1 Fig). We chose to show this analysis as it is the most straightforward one.

The difference shown in Fig 6C could stem from two different effects—a larger drift in the spontaneous activity due to stimulation of the network, and a specific drift of the stimulated vs. the non-stimulated patterns within the stimulated networks. To dissociate the two, we now only consider the stimulated networks. For each network, we chose the clusters that were

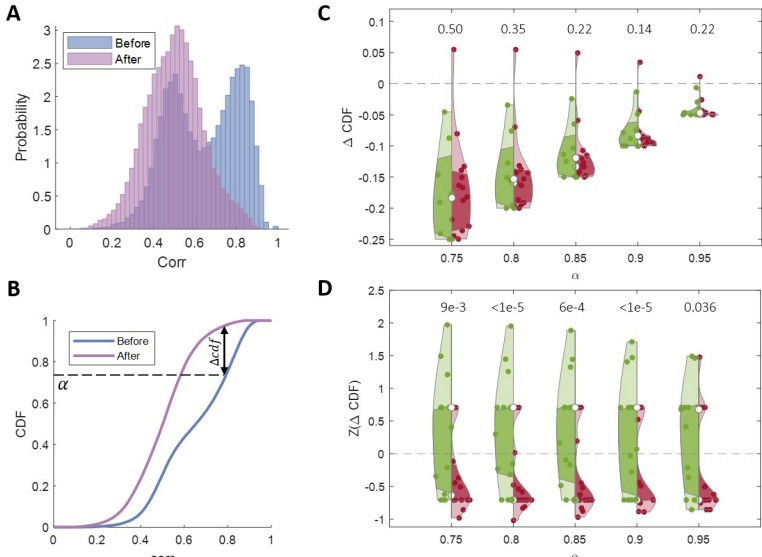

**Fig 6. Changes in the spontaneous activity. (A)** 2D correlation values between all spontaneous bursts and 25 bursts from cluster 5, before (blue—high correlations) and after (purple—low correlations). **(B)** CDFs of the 2 distributions shown in (A). We quantify the effect by considering the most correlated bursts—the top $1 - \alpha$ percent. This defines a correlation threshold, from which we can measure which percent of the bursts cross this threshold after stimulation. We measure this difference for $\alpha \in [0.75, 0.95]$. **(C)** Existence of effect—stimulation vs. control. Statistics across 11 stimulation experiments (Table 1) and 5 control experiments (Table 2). The violins represent $\Delta CDF$ values in stimulation experiments (red) and in control experiments (green) for a range of $\alpha$ values. The numbers above each pair of violins represent the p-value of the hypothesis that the effect in the stimulation experiments is larger than in the control experiments. **(D)** Specificity of the effect—measuring the effect in the spontaneous vocabulary. The violins represent the z-scored $\Delta CDF$ values for the stimulated clusters (red) and for the non-stimulated clusters (green), see S2 Fig for raw $\Delta CDF$ values. The numbers above each pair of violins represent the p-value of the hypothesis that the effect in the stimulated clusters is larger than in the non-stimulated clusters. The same analysis was done for the control experiments as well (see S3 Fig).

robustly evoked by stimulation (Fig 5, see Methods) and calculated $\Delta CDF$. We additionally chose the same number of non-stimulated clusters (see Methods) for each network and repeated the same analysis. Networks are expected to differ not only in their correlation thresholds but also in their baseline drift rates. We therefore z-scored the $\Delta CDF$ values within each network before combining them across networks (Fig 6D). We can see that, on average, $\Delta CDF$ is negative for the stimulated patterns, while positive for the non-stimulated ones. This indicates that indeed the larger drift results from specifically stimulating these patterns.

One simple possible explanation for this effect is that the stimulated pathways were "damaged" such that the network is no longer able to generate these patterns. Such damage could arise, for instance, from a homeostatic increase in the firing threshold of highly active neurons [23]. Analyzing the evoked responses, however, shows the opposite is true. Throughout the 10-hour stimulation, not only that the network continues to generate these evoked responses, but they also get more robust with time. This can be visually appreciated by looking at the center-of-mass projections of one evoked response (site 17, Fig 7A), in which later responses are more tightly concentrated in space. We quantify this effect by measuring the variance between the evoked responses in windows of 30 minutes (Fig 7B), and repeated the analysis across all networks (Fig 7C). In other words, these pathways remained accessible via stimulation but became almost unreachable spontaneously. One can say that there is now a new association between the stimulated attractors and the occurrence of the specific stimulation that generates an evoked convergence to them.

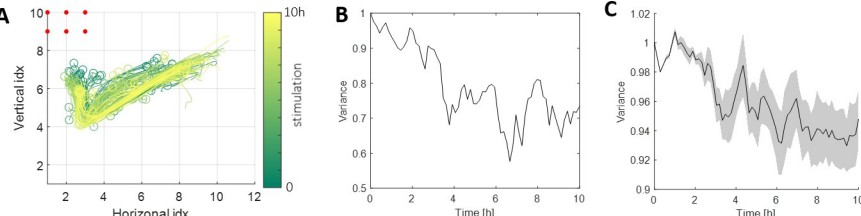

**Fig 7. Evoked responses become more robust. (A)** Center of mass trajectories of the responses to site 17 throughout the 10-hour stimulation (color: 0 (green) to 10 (yellow) hours). The 6 red dots denote the specific stimulating electrodes. **(B)** Variability between the evoked responses in (A) throughout the 10-hour stimulation period. The variance is calculated in windows of 30 minutes throughout stimulation. In each window, we measure the deviation from the window's mean response (Euclidean distance). The plot is normalized by the variance at the first window. **(C)** Mean and variance of the variability between the responses for each stimulation site (statistics across all 11 experiments (Table 1)).

## Mechanism

These effects raise many interesting questions—What causes this phenomenon? What is the mechanism behind these vocabulary changes? How do the background dynamics change to support such changes?

We imagine these two effects in the following way: Each network has a set of multiple discrete attractors that can be reached spontaneously, while some of them can also be reached through electrical stimulation. We show that, throughout stimulation, the evoked responses become more robust with time—consistent with the basin of attraction becoming steeper on one side. On the other hand, the same attractors become much less accessible spontaneously—consistent with another side of the basin becoming flatter. One can imagine digging in the energy landscape and piling the dirt onto the other side.

In order to verify this hypothesis, we need to map the basin of attraction before and after stimulation. We do this via the set of initial states of bursts. Specifically, we ask whether the same set of initial states will lead to the same set of bursts. We define such a candidate set by considering the initial states of one of the stimulated clusters (see Methods). We can now follow all the bursts that originate from this area. Before stimulation, these bursts are similar to one another (the peak at large correlation values in Fig 8A left, blue). After the stimulation, however, the bursts originating from the same area are much more variable (Fig 8A left, purple). To quantify this difference, we once again calculate $\Delta CDF$ as shown in Fig 6B. Repeating the analysis on initial states stemming from a non-stimulated cluster shows a smaller effect (Fig 8A, right). We pool the data from all networks using z-scores of this value, showing a trend for the stimulated patterns to be more disrupted (Fig 8B).

We can visualize the change in the dynamics by looking at all the bursts associated with a single cluster—evoked and spontaneous, before and after the stimulation. Using nonlinear dimensionality reduction, we can see that for the non-stimulated patterns (Fig 8C right, green frame), similar initial conditions lead to similar bursts. For the stimulated cluster (left, red frame), however, this is only true before stimulation (narrow distribution of blue trajectories), and not after (purple trajectories).

## Discussion

In this work, we analyzed the spontaneous activity of cultured neural networks. We showed that each such network has a finite repertoire of bursts that function as discrete attractors. Based on these dynamics, we were able to create a vocabulary of spatiotemporal patterns that

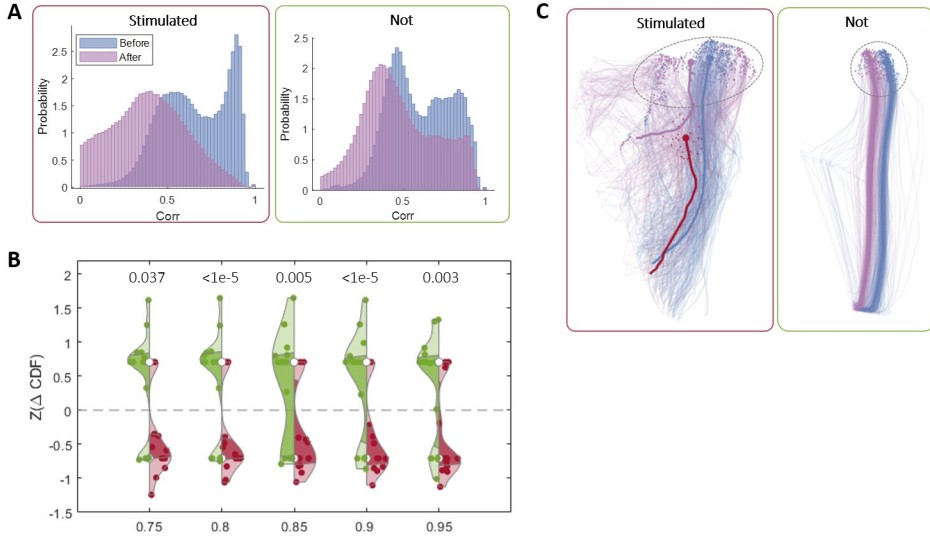

**Fig 8. Mechanistic explanation. (A)** The probability distribution of the 2D correlations between pairs of bursts with initial states similar to the ones of the stimulated patterns (red frame) and not similar to the stimulated patterns (green frame), before stimulation (blue), and after (purple). The difference between the two is measured by $\Delta CDF$ as shown in Fig 6B. **(B)** Statistics across 11 stimulation experiments (Table 1). The violins represent $\Delta CDF$ values in stimulation clusters (red) and in non-stimulated clusters (green) for a range of $\alpha$ values. The numbers above each pair of violins represent the p-value of the hypothesis that the effect in the stimulation clusters is larger than in the non-stimulated clusters. The same analysis was done for the control experiments as well—see S4 Fig. **(C)** Trajectories in a non-linearly reduced 2D space (using UMAP) of one stimulated cluster (left, red frame) and one non-stimulated cluster (right, green frame), before (blue) and after (purple) stimulation. The dashed circles denote the area of initial states for each cluster.

describe the spontaneous dynamical space of the network. We showed that these attractors are accessible not only spontaneously, but also using electrical stimulation—we were able to find stimulation sites that generated robust and coherent evoked responses similar to the ones in the spontaneous vocabulary.

In order to answer questions regarding the plasticity of the vocabulary, we used electrical stimulation to force the network's dynamics to visit specific attractors repeatedly. We find that the targeted attractors are eliminated from the spontaneous vocabulary, while they are robustly evoked by the electrical stimulation. This seemingly paradoxical finding can be explained by a Hebbian-like strengthening of specific pathways into the attractors, at the expense of homeo-static-like weakening of non-evoked pathways into the same attractors.

Synchronized bursts are routinely observed in neural cultures and have been suggested to be a barrier to plasticity [24]. Therefore, several attempts have been made to suppress them in order to allow plasticity [25, 26]. Our work suggests that these synchronized bursts can also be informative, and serve as objects that advance the study of plasticity. In this work, we learned the network's dynamical structure and used it as a tool to shape the dynamics in specific directions. This is similar to the concept of learning within the intrinsic manifold presented in [27] which suggests that working within the constraints imposed by the underlying neural circuitry can make the learning process significantly easier and more accessible.

To our knowledge, this work provides the first direct evidence for the plasticity of multiple attractors in a biological neural network. The plasticity principles we describe improve our understanding of how attractors in a biological system evolve.

## Methods and materials

### Cell culture

Cortical neurons were obtained from newborn rats within 24h after birth as described in [28]. The neurons were plated directly onto multielectrode arrays (MEAs) and allowed to develop mature networks over a time period of 18–21 days. The number of neurons in a typical network is in the order of $10^6$. The preparations were bathed in Minimal Essential Medium (MEM) supplemented with NuSerum (10%), L-Glutamine (2mM), glucose (20mM), and insulin (25mg/l), and maintained in an atmosphere of 37°C, 5% CO2 and 95% air in an incubator. Starting a week after preparation, half of the medium was replaced every 2 days with a fresh medium similar to the one described above excluding the NuSerum and with lower concentrations of L-Glutamine (0.5mM) and 2% B-27 supplement.

During recordings and stimulation, the cultures were removed from the incubator, but still maintained in an atmosphere of 37°C, 5% CO2, and 95% air. The dish was perfused at a constant ultra-slow rate of 2.5 ml/day by a custom-built perfusion system.

### Experimental system

Network activity was recorded and stimulated through a commercial 120-channel headstage (MEA2100, MCS). The 120 30$\mu$m diameter electrodes are arranged in a 12x10 array, spaced 1mm vertically and 1.5mm horizontally. Data acquisition was performed using Multi Channel Suite. All data were stored as threshold crossing events, with the threshold set to 5$\sigma$, where $\sigma$ is the standard deviation of the entire voltage trace. All thresholds were separately defined for each of the recording electrodes before beginning the experiment protocol. Each protocol was started only after the culture rested on the system for each least 30 minutes during which we verified by visual inspection that indeed threshold-crossings correspond to clear spike events.

**Stimulation profile**: As described in the text, 6 electrodes were selected for stimulation at each stimulation site. Biphasic voltage pulses of plus and minus 700$mV$ lasting 400$\mu$sec, 200$\mu$sec respectively for each phase were activated through all 6 electrodes simultaneously.

### Data processing

Threshold crossings (5$\sigma$) yield discrete time stamps of spike events from 120 extra-cellular electrodes. Each electrode records spike events from a number of adjacent or distant neurons (multi-unit recordings). We smooth (using a Gaussian kernel, $\sigma$ = 2msec) and bin the data (bin size is 5 msec) to get 120 continuous time series, in 5 msec resolution (Fig 9).

Following stimulation, we noticed two effects: An electrical artifact lasting less than 2 msec, and spanning many electrodes; In addition, the 6 stimulating electrodes exhibited modified waveforms for roughly 200 msec following stimulation. These were processed as many spiking events by the system, with much shorter inter-spike-intervals than for the other electrodes (5 ± 1 vs. 12 ± 3). To mitigate stimulation artifacts and to only consider events that were well isolated single spikes, we excluded the first 5 msec for all electrodes, and the entire response for the stimulated electrodes.

### Network burst extraction

We detected all spontaneous bursting events using threshold crossing with the threshold set to 4$\sigma$, where $\sigma$ is the standard deviation of the overall activity—determined separately for each network and each epoch (before and after stimulation). Using these events as a reference, we defined the duration of each event by searching for the first crossings of 0.5$\sigma$ before (starting point) and after (ending point) the 4$\sigma$ timestamp. The typical duration of these spontaneous

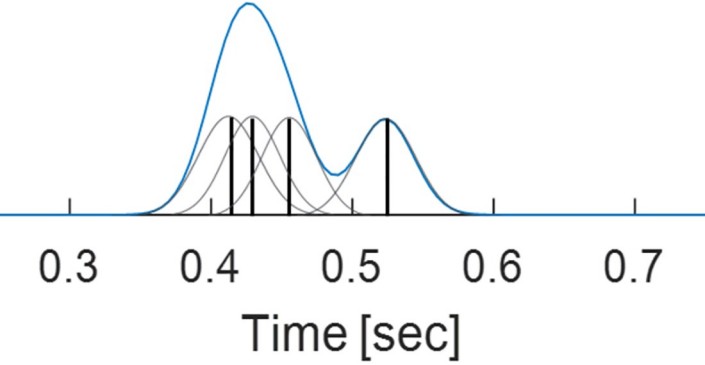

**Fig 9. Continuous smoothed signal.** Each spike event (a discrete timestamp, black vertical lines) is smoothed using a Gaussian kernel (grey). Then, we bin the data (5 msec) and sum the over all signal in each bin. The result is a continuous smoothed signal (blue) for each of the 120 electrodes.

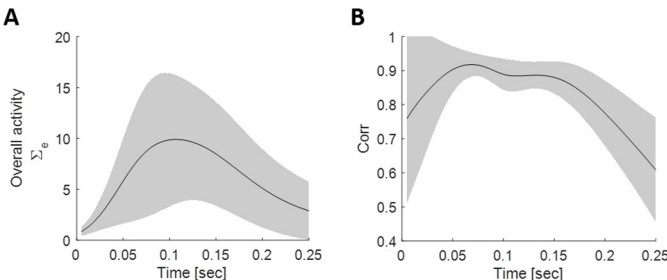

**Fig 10. Network bursts—Statistics. (A)** Mean (black line) and standard deviation (grey shading) of the overall activity (sum over all 120 electrodes) across all network bursts in a single network. Only bursts that last at least 250 msec are considered. **(B)** Mean (black line) and standard deviation (grey shading) of the pair-wise correlation values across time between similar bursts (2D corr $>$ 0.85). Only bursts that last at least 250 msec are considered.

bursting events is 100 to 250 msec. In our analyses, we focus specifically on the first 100 msec of these busts, as this time-frame tends to exhibit the greatest variability among them (Fig 10).

## Clustering method

When clustering the spontaneous activity, we relied on the observation that similar initial conditions lead to similar patterns (bimodal distribution in Fig 2A). There are many possible metrics for comparing the spatiotemporal activity patterns. To capture different aspects of the bursts, we used 3 different metrics to measure the similarity between them. In each case, we consider two bursts $X_1, X_2 \in R^{N \times T}$, where $N = 120$ and $T = 20$ (100msec in 5msec bins).

1. **2D correlation**

   The correlation coefficient between 2 bursts $X_1, X_2$ is computed in the following way:

$$corr2(X_1, X_2) = \frac{\Sigma_t \Sigma_n (X_{1_{tn}} - \bar{X}_1)(X_{2_{tn}} - \bar{X}_2)}{\sqrt{\left(\Sigma_t \Sigma_n (X_{1_{tn}} - \bar{X}_1)^2\right)\left(\Sigma_t \Sigma_n (X_{2_{tn}} - \bar{X}_2)^2\right)}}$$

   where $\bar{X}_1$ and $\bar{X}_2$ are the average of $X_1$ and $X_2$ over both dimensions (electrodes and time).

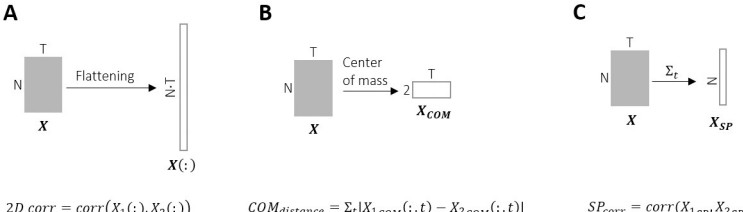

**Fig 11. Evaluation metrics. (A)** 2D correlation: correlation coefficient between the two flattened bursts. **(B)** Center of mass (COM) distance: The sum of all Euclidean distances between the COM of two bursts. **(C)** Spatial correlation: correlation coefficient between the spatial profiles of two bursts.

This is equivalent to flattening both matrices $X_1$ and $X_2$ and subsequently computing the Pearson correlation. See Fig 11A.

2. **Euclidean distance between the center of mass of trajectories**
   We compute the center of mass (COM) of a burst as a weighted average of the activity from all 120 electrodes. Namely, each electrode $n$ has coordinates $x_n \in R^2$ on the MEA. The 2D trajectory of the center of mass of burst $X_1$, denoted $X_{1_{COM}} \in R^{2 \times T}$ is then:

$$X_{1_{COM}} = \sum_n x_n X_{1_{tn}}$$

The Euclidean distance between 2 such trajectories is computed as the norm of the difference between the two across time: $\Sigma_t |X_{1_{COM}} - X_{2_{COM}}|$. See Fig 11B.

3. **Correlation of spatial profiles**
   The identity of the active electrodes is used to define this metric. The spatial profile of a burst $A_t n$ is defined as a vector of length $N$ capturing the overall activity of each electrode throughout the burst: $SPX_1 = \Sigma_t X_1$. The correlation coefficient between $SPX_1$ and $SPX_2$ is used. See Fig 11C.

The actual range of values for these 3 metrics varies between networks. In order to obtain measures that are more invariant, we rely on the bimodal distributions of the initial state and the full burst similarity shown in Fig 2A, but now extended to all three metrics in Fig 12A–12C. For each network and each metric, we defined two thresholds (shown in dashed white

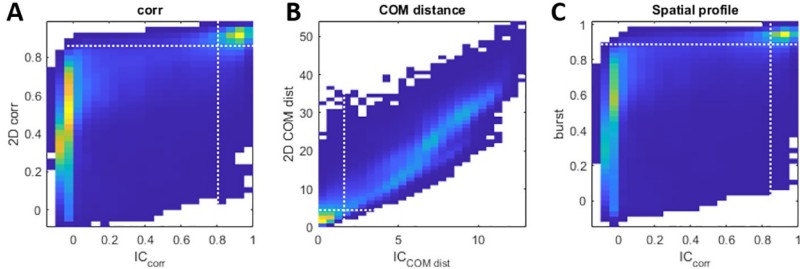

**Fig 12. Dynamics-based clustering. (A)** For each pair of spontaneous bursts we compute the 2D correlation value and the correlation between the initial states, then we plot the probability density of the values of the first as a function of the second. The dashed white lines represent the thresholds that define the pairs of bursts that converge to the same attractor (right upper square). **(B)** Same as in A, using the center of mass (COM) distance metric. **(C)** Same as in A, using the spatiotemporal profile metric.

lines) in order to distinguish between pairs of bursts that converge to the same attractor (close initial states and similar bursts) and pairs of bursts that converge to different attractors.

Based on this distinction we can build a graph for each of the three metrics: each node is a burst; two nodes are connected if they cross both the initial condition threshold and the metric threshold. Then, we sum these 3 non-directed graphs into a single similarity graph $S$ with edges valued 0–3, where $S_{i,j} = 0$ means that burst $i$ and burst $j$ are not connected and therefore not similar.

We perform spectral clustering [18] using the MATLAB function *spectralcluster* with the following specifications:

- We use the normalized symmetric Laplacian matrix $L_s = D_g^{-1/2} L D_g^{-1/2}$ where $D_g$ is the diagonal matrix obtained by summing the rows of the similarity matrix $S$.

- We use k-medoids as the clustering method.

We only consider clusters that capture at least 2% of spontaneous bursts, which accounts for the vast majority of bursts (see Tables 1 and 2, "percent explained").

### Selecting stimulation sites

The experiment protocol comprises several phases, one of which is the probe phase, lasting 1.75 hours. During this phase, we record the network's evoked responses from a predefined set of 20 stimulation sites. Once the probe phase is completed, we need to identify the top 3 stimulation sites that elicit the most robust responses in order to proceed with a 10-hour stimulation using only these 3 selected sites. To achieve this, we conduct a brief analysis, taking up to 30 minutes, during which we assess the network's evoked responses to all 20 stimulation sites. Our goal is to identify three stimulation sites that consistently produce robust and coherent responses. Furthermore, we aim to select three sites that are as spatially distant from each other as possible, and whose evoked responses differ significantly from one another.

To aid in this selection process, we employ several visualization techniques:

- Center of mass trajectories: we view all 30 evoked responses for each of the 20 stimulation sites in order to get a qualitative assessment of the robustness of each response (Fig 13A).

- Pair-wise correlation values: We examine these values both within each site and collectively across all sites to assess the similarity among all 30 repetitions and dissimilarity from all the other sites (Fig 13B). We wish to identify those with a concentrated distribution of high within-correlation values (indicated in orange), minimizing any overlap with the across-site distribution (depicted in blue).

- A dendrogram constructed from pair-wise correlation values across all stimulation sites: serves as a tool before the final selection. We wish to choose three distinct responses, each belonging to separate branches within the dendrogram (Fig 13C)

### Relating evoked responses to spontaneous clusters

In order to measure the existence of the evoked responses in the spontaneous vocabulary of the network, we wish to relate them to specific spontaneous clusters (shown in Fig 3E). To this end, we define a cluster as *explaining* a specific repetition of a specific stimulation site. Comparing evoked responses to spontaneous bursts entails two difficulties:

- Stimulating electrodes in the evoked responses exhibit a qualitatively different signal.

- Different initial states for spontaneous and evoked bursts (Fig 3F)

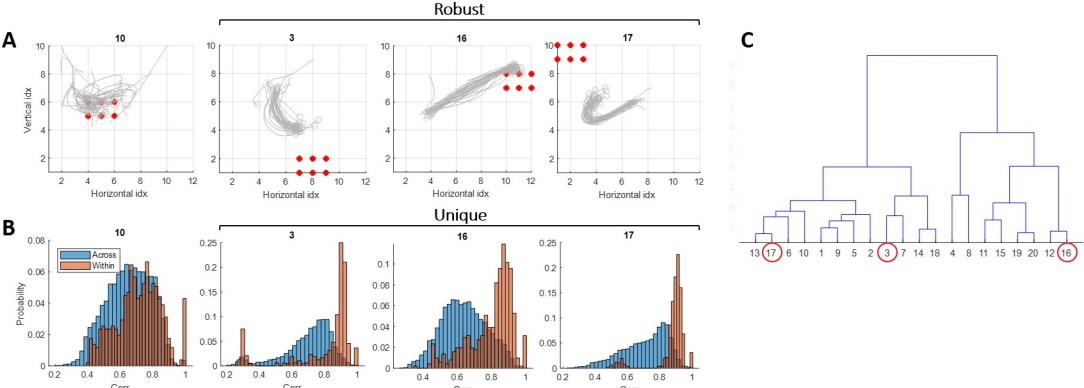

**Fig 13. Selecting stimulation sites. (A) Robustness.** Visualization of the evoked responses to the selected sites 3,16,17 and site 10 serving as an example of one that did not meet the selection criteria. Each site is represented by the center of mass for all 30 responses. Red dots indicate the 6 stimulating electrodes in each case. Sites 3, 16, and 17 display robust and coherent responses, in contrast to site 10. **(B) Uniqueness.** Comparing the evoked responses—within each site and across sites. Here we show the pair-wise correlation values within the evoked responses to a given site (orange) and the pair-wise correlation values between the evoked responses to this site and the evoked responses to all other sites (blue). The pairwise correlation values within the selected sites (3,16,17) exhibit a narrow distribution centered around 0.9, while the values for site 10 are widely dispersed, with a mean around 0.7. Furthermore, within the selected sites, the distribution is distinctly different from the pair-wise correlation values across sites. In contrast, for site 10, these two distributions exhibit a significant overlap. **(C)** Selecting sites that generate different evoked responses. Here we show a dendrogram that was constructed from the pair-wise correlation values across all stimulation sites. In the case presented, we selected sites 3, 16, and 17 (red circles). In addition to their robustness and uniqueness (shown in A and B), they were selected for their clear separation into distinct branches within the dendrogram.

We thus exclude the stimulating electrodes before computing the 2D correlation between evoked and spontaneous bursts. The threshold for similarity is expected to be lower than that used to compare between two spontaneous bursts. We thus use an adjusted $\theta$ value (horizontal dashed line in Fig 2A) reduced by some factor $f_{th}$.

During the probe phase, we stimulate the network in 20 different sites, 30 times each. For each such single response, we perform the following:

- Compute the pairwise 2D corr to all the spontaneous bursts

- If $2Dcorr > f_{th}\theta$, this pair is considered similar

- We only consider single responses that are similar to at least $m$ percent of all the spontaneous bursts.

- For each such response, we say that it is *explained* by the cluster for which most of the similar spontaneous events belonged to.

The results shown in Figs 3E and 5 are based on $f_{th} = 0.88$ and $m = 0.5\%$. The following parameter combinations yielded qualitatively the same results: $(f_{th}, m) = (0.88, 1\%), (0.85, 0.5\%), (0.85, 1\%), (0.9, 0.5\%), (0.9, 1\%)$.

## Stimulated and non-stimulated clusters

In our analysis, we define for each network a set of clusters that are similar to the three evoked responses ("stimulated clusters") and a set of non-stimulated clusters. The definition of these two sets relies on the analysis shown in Fig 5. For each stimulation site in the green area, we defined a stimulated cluster as the one which explains most of the evoked responses to a given

stimulation site. The non-stimulated clusters were all the clusters that explained *none* of the evoked responses to *all* 3 stimulation sites.

## Measuring the difference between CDFs

We quantified the changes in the existence of patterns using the cumulative probability distribution of correlation values between spontaneous bursts. The actual correlation values differ between networks and between patterns. Therefore, we use a pattern-specific threshold as a reference. Using this threshold, we can calculate the change in the existence of patterns before and after stimulation in the following way:

$$\Delta CDF = \alpha - CDF_{after}(CDF_{before} = \alpha)$$

Since we care about changes in high values of correlation, we measure this difference for a range of values: $\alpha \in [0.75, 0.95]$.

This way we are able to measure the percentage of highly correlated spontaneous bursts to a given pattern, before and after stimulation, in a pattern-specific manner.

## Statistics across networks

Our data set consists of 11 stimulation experiments (Table 1) and 5 control experiments (in which there was no stimulation during the 10 hours; Table 2). The tables summarize some of the activity characteristics of each culture: MEA serial number, cell preparation date, the number of days elapsed between the preparation date and the experimental protocol initiation, the number of spontaneous bursts per hour (before and after stimulation), the number of clusters (dictionary size; before and after stimulation) and the overall percentage of network bursts that were clustered (clusters containing less than 2% of all spontaneous busts are discarded), and number of stimulated patterns (out of the selected 3) that that were shown, post-hoc, to be robust and existent (Fig 5).

## Success rate & probe as a criterion to proceed

The total number of cell preparations done in this study is about 100. A large number of them did not develop well enough (due to contamination events, low density of cells, and other reasons related to the maintenance atmosphere) and therefore were cleaned at early ages. The ones that matured successfully were transferred to the experimental system. We performed 17 stimulation experiments and 14 control experiments on cultures between the ages of 18–21 days. Some of these experiments are not part of the results presented in this paper due to low responsiveness to the 20 stimulation sites.

After the first 4-hour recording of spontaneous activity, there are 1.75 hours in which we probe the culture in 20 different sites, repeatedly. After this probing, we do a short analysis in which we pick the 3 stimulation sites which generated the most robust and coherent responses, then we continue the protocol as shown in Fig 4. In some of the cultures, there were no such responses at all; In these cases, we stopped the experiment right after the probing. The percentage of experiments (stimulation and control) that were completed (responded the at least 3 distinct stimulation sites robustly) is about 50%.

The decision of whether to continue an experiment after the probing stage was not based on a clear-cut condition, but on evaluation based on several figures (Fig 13) aiming to evaluate the robustness of the responses. If there were less than 3 robust and distinct responses, or when there were very low activity levels (low number of participating electrodes), we stopped the protocol.

## Supporting information

**S1 Fig. Measuring the effect using the probe evoked responses.**
(TIF)

**S2 Fig. Specificity of the effect—Raw values.**
(TIF)

**S3 Fig. Specificity of the effect—Control experiments.**
(TIF)

**S4 Fig. Mechanism—Control experiments.**
(TIF)

**S1 Text. Measuring the effect using the probe evoked responses.**
(PDF)

## Acknowledgments

We thank Shimon Marom for many discussions and comments along the project. We thank Noam Ziv for useful comments on the manuscript, as well as Tamar Galateanu and Leonid Odesski for their technical support.

## Author Contributions

**Conceptualization:** Chen Beer, Omri Barak.

**Funding acquisition:** Omri Barak.

**Investigation:** Chen Beer, Omri Barak.

**Methodology:** Chen Beer, Omri Barak.

**Software:** Chen Beer.

**Supervision:** Omri Barak.

**Validation:** Chen Beer.

**Visualization:** Chen Beer.

**Writing – original draft:** Chen Beer, Omri Barak.

**Writing – review & editing:** Chen Beer, Omri Barak.

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
