## [Decision Letter · Decision Letter 0]

16 Aug 2023

Dear Mrs. Beer,

Thank you very much for submitting your manuscript "Revealing and reshaping attractor dynamics in large networks of cortical neurons" for consideration at PLOS Computational Biology.

As with all papers reviewed by the journal, your manuscript was reviewed by members of the editorial board and by three independent reviewers. In light of the reviews (below this email), we would like to invite the resubmission of a significantly-revised version that takes into account the reviewers' comments. In particular, we would ask you to clarify the description of the analysis, and to address the suggestion by reviewer 1 to check how the spatial correlations behave over longer time intervals.

We cannot make any decision about publication until we have seen the revised manuscript and your response to the reviewers' comments. Your revised manuscript is also likely to be sent to reviewers for further evaluation.

Sincerely,

Matthias Helge Hennig, Ph.D.

Academic Editor

PLOS Computational Biology

Lyle Graham

Section Editor

PLOS Computational Biology

Reviewer's Responses to Questions

**Comments to the Authors:**

Reviewer #1: In this manuscript by Beer and Barak, the authors use dissociated cortical cultures grown on a 120-electrode array to investigate attractor dynamics before and after electrical stimulation. While there is an extensive literature on how synapses change after electrical stimulation (LTP, STDP, etc.), there is relatively little work on how networks of neurons rewire themselves in response to such stimulation. Previous works showed that the strength of some pathways went up, while others went down, but there was not a framework describing how these changes would occur. Here the authors attempt to offer some clear results that may guide future work.

Briefly, they find that each culture has a small but diverse repertoire of activity patterns that spontaneously occur. After appropriately targeted stimulation, some of these activity patterns appear less often spontaneously, but paradoxically are more reliably evoked after stimulation. Another interesting feature of each repeating activity pattern is that individual instances of it become more similar over time, as they each unfold in about 200 ms. This leads the authors to conclude that these activity patterns are attractors that can be evoked from multiple different configurations and yet still converge to nearly the same final configuration. Such attractors are widely invoked in the artificial neural network literature (e.g., Hopfield networks) but experimental evidence for them is rare.

This result is very interesting and relatively novel (there has been some work by the Buonomano group in this general area), so I think this work would be of broad interest to the readers of PLoS Computational Biology. The explanation offered for this result is plausible (Hebbian plasticity) and fits into the prevailing picture of how networks learn. In this respect, the manuscript could be seen as clear proof of attractors that have long been hypothesized to exist. Previous work in this area (e.g., Cossart, Aronov and Yuste, 2003; Segev, Baruchi, Hulata and Ben Jacob, 2004; Beggs and Plenz, 2004) showed that repeating patterns of activity existed, but failed to demonstrate attractive dynamics.

In what follows, I will offer suggestions for the authors to consider.

Presentation:

It took me a while to read and re-read your manuscript, even though I work in this area. My main difficulty was that several of your explanations were very terse. Sometimes details were omitted, and other times labels or graphs could have been explained better. I think you should have outside people read it carefully and indicate parts where they do not understand clearly what you are doing. I will give some specific examples below:

In Figure 2, you should show to the reader the correspondence between the attractors plotted in panel C and those in the embedding in panel D. There are nine attractors and nine trajectories in the embedding. I can probably guess which ones correspond between C and D, but you could also label them to help the reader. Regarding panel D, I looked for a citation or brief explanation of UMAP but could not find one. While the paper is very widely cited, you should still give readers some indication of what you are doing. No colorbar is shown for figure 2 D, only contour plots without explanation. Again, while I can probably guess or work it out, you could also help the reader here. In panel C, are the circles the beginnings of the trajectories? I did not see that indicated until much later in the caption for figure 8, panel C.

Reference 10 seems like a non-sequitur.

In the methods section, under Data processing, you say that you bin the data in bins of 5 ms. Do you ever get more than one spike in a single bin? If so, how do you handle it? It could be that you just give a 1 if there are two spikes there, or you could give a 2, counting both spikes. How often did you get multi-unit activity (MUA)? How often did you get clear single spikes? Were there any local field potentials that could have produced threshold crossings? If so, did you discard these or count them as activity at the electrodes? I could not find where these issues were addressed.

Under Clustering Method, your explanations are unclear to me. For example, in describing the spatiotemporal correlation, are you performing vector subtraction element by element in the numerator of the equation? That is my guess, but you could be clearer. In describing the Correlation of spatial profiles, are you taking the amplitude to be the sum of the active electrodes at each time bin? Please explain. Perhaps a toy example illustrating your methods could help here.

On line 86, you mention that you use spectral clustering methods to group the trajectories with the most similarity into your vocabulary. While I have used spectral clustering methods, I think you should describe to the reader a bit more how you did this. Did you use a greedy algorithm, or did you anneal for your solutions? This is a computational biology journal, so I think you should describe a bit more about the steps in your computations.

I found Figure 9 D to be a challenge. You offer only a brief explanation of what is presented there, and I cannot see the axes. Are there directed connections there or are all the connections undirected? Why does the plot seem to look like it has a void in the middle? Does this tell us anything? What important point of your paper is it supporting? This was not obvious to me.

In some of the figures, you use COM to presumably describe Center of Mass. Please just spell it out in the captions so people do not have to guess what it means.

Scientific issues:

One of my main concerns is over whether you actually have attractors or not. I think I get the logic behind figure 2 B, where the variance declines over time. I really like that, but I am wondering about what happens if you plot this longer than 0.15 seconds. Does the variance go up again after that? Why stop at 150 ms? I would like to see if they are becoming more similar over time merely because more electrodes are getting activated. If most of the array is just becoming activated, then of course nearly any starting configuration will look more similar over time. There would be just one large attractor like what you might see in a seizure. Please convince me that this is not the case.

Any time a threshold is chosen, one may wonder how sensitively the results depend on its exact value. If you take the threshold theta for clustering correlations described in the caption to figure 2A and change it by say + or – 20%, do you still get qualitatively the same results? This would give me some more confidence if you could show it.

When discussing the artifacts caused by electrical stimulation (line 120), you did not mention doing a TTX subtraction. As you probably know, washing in TTX will show you what portion of the artifact was caused by capacitance and resistive effects that do not depend on sodium channels. This signal can later be subtracted from the full recorded waveform to reveal only that portion that was caused by neuronal firing. Of course, you only want to wash in the TTX at the very end of the experiment when you are no longer recording activity. TTX can be washed out, but it takes some time.

A suggestion on the interpretation of your results: I can see how Hebbian plasticity could account for the fact that the frequently stimulated attractors become more responsive to stimulation over time. But the observation that they are less often spontaneously evoked could be explained by homeostasis. I do not think you mentioned homeostasis, but you should probably consider it. See, for example (Liu, Seay and Buonomano, 2023).

Reviewer #2: The manuscript is devoted to the study of evoked activity in cultured neural networks. First, the authors analyze synchronized bursting events structure. The idea behind this analysis is that the network has an attractor dynamic which converges into stereotypic activity across all recording sites. Then, they move to evoke these events using repeated stimulation of the network. They show that electrically stimulating specific attractor eliminates it from the spontaneous vocabulary, while it is still robustly evoked by the electrical stimulation. Overall, the work is interesting, and the analysis is solid.

My only major comments, it is not clear what is new in this study. The structure of synchronized bursting events was analyzed by Ben-Jacob et al, and the effect of stimulus on the network dynamics was demonstrated by Maron et al. It would be nice if the authors explain explicitly what is novel here. In addition, the authors can do a better job in citing prior art in the analysis of synchronized bursting events.

Minor comments:

Figure 1A: maybe scale bar missing?

Figure 1B right panel, what are axis labels?

Figure 2 caption: what is upside down ‘?’?, panel C, what are the axis?

Figures 3, 7, 8, all panels should have labels and units.

Figure 9D: I do not understand panel D.

Reviewer #3: The authors claim that 1) they found attractors in spontaneous activity of cultured neural networks, 2) these attractors can be evoked by electrical stimulation of said networks and 3) prolonged stimulation of those attractors removes them from the spontaneous activity, while stimulation can still trigger them. Activity occurred in bursts, as seen before for cultures. Events were defined as bursts that cross a certain threshold in terms of population participation and are analyzed for 100 ms after their start. Figure 2 represents an impressive approach to experimentally identify attractor dynamics and categorize patterns within spontaneous activity. From there, the authors convincingly demonstrate that repeated stimulation can trigger robustly intrinsic patterns which then are less reliably recalled spontaneously by the network.

Overall, the study is very interesting, methods utilized are of very high quality, and utilized in a convincing manner for the questions proposed. Conclusions are properly supported by their results and have relevant consequences in a wide area of systems neuroscience, raising important follow up questions that should be further explored. Overall, the paper is written very succinctly, to the point that numerous details of the results are neither mentioned nor discussed. My comments below should encourage the authors to elaborate in more detail some of the aspects of this otherwise promising work.

Comments:

1) In Fig. 2A, we see the density plot for correlation of overall burst vs initial condition. It can be observed a large fraction of burst pairs have close to zero initial correlation but large overall burst correlation. Any ideas where that comes from?

2) In Fig. 2D, we see a low-dimensional representation of the trajectories for all clusters extracted from a single (?) culture. There it becomes clear that all bursts tend to move towards the middle of the MEA. Is there a reason for this? Perhaps a bias on connection densities? It also can be noted that bursts tend to not start at the center, do we know why that is the case?

3) Stimulation occurred in a predetermined sequence that is also mostly spatially contiguous (site 1 is close to site 2, which is close to site 3, etc.). Could that have consequences in the responses observed? For instance, if by stimulating site 1, certain neurons become depleted and can’t participate when site 2 is stimulated next, that could narrow down possible trajectories in response to site 2 stimulation. Furthermore, stimulation happened at an interval of 10 s. It seems bursting in spontaneous activity occurred at an average interval of ~14 s (based on numbers given by the authors). Could the shorter interval for stimulation also be a factor here? Maybe the data obtained from the repeated stimulation of specific sites can be used to answer this question.

4) In Fig. 3B, we see robust and coherent responses to stimulation site 17, but it’s intriguing that the initial point in those trajectories is so far removed from the stimulation sites? Any ideas why this is the case? Also stimulation site 10 doesn’t seem to yield coherent responses. Could it be linked to the fact that this site is close to the center (maybe related to the question in point 2 above)?

5) In Fig. 4, we see a very small fraction of responses that are very robust (to the right of the vertical dashed line) but couldn’t be “explained” by the spontaneous vocabulary. Did the authors further investigate these responses? It would be interesting to understand what conditions give rise to those that are not present during spontaneous activity.

6) Since the authors do have control experiments in which no stimulation was performed, it would be important to obtain some more statistics on changes to existing clusters after that 10 h period, so the effects of stimulation are taken with more context. For example, one can see in table 2 that the number of clusters change for all cultures. Are these mostly from a few clusters being added/removed from the previous set, or are we seeing completely new set of clusters after 10 h? How similar are the clusters before and after?

7) Finally, I would suggest being a little more detailed on the definitions necessary to understand the stimulation part (Figs. 6 to 8). While they can all be found in the Methods section, I believe it would help the reader to follow the results more clearly if they were explained in a little more detail.

**Have the authors made all data and (if applicable) computational code underlying the findings in their manuscript fully available?**

Reviewer #1: Yes

Reviewer #2: Yes

Reviewer #3: **No: **a github link has been provided but not data or code has been deposited yet.

PLOS authors have the option to publish the peer review history of their article (what does this mean?). If published, this will include your full peer review and any attached files.

Reviewer #1: **Yes: **John M. Beggs

Reviewer #2: No

Reviewer #3: **Yes: **Dietmar Plenz
---

## [Decision Letter · Decision Letter 1]

22 Dec 2023

Dear Dr Beer,

We are pleased to inform you that your manuscript 'Revealing and reshaping attractor dynamics in large networks of cortical neurons' has been provisionally accepted for publication in PLOS Computational Biology.

Best regards,

Matthias Helge Hennig, Ph.D.

Academic Editor

PLOS Computational Biology

Lyle Graham

Section Editor

PLOS Computational Biology

Reviewer's Responses to Questions

**Comments to the Authors:**

Reviewer #1: I think the authors have done a good job of responding to my queries. I am satisfied with the manuscript as it currently stands. It provides interesting new information on the variety and dynamics of attractors in cultures of dissociated neurons.

Reviewer #2: The new version is improved.

**Have the authors made all data and (if applicable) computational code underlying the findings in their manuscript fully available?**

Reviewer #1: Yes

Reviewer #2: Yes

PLOS authors have the option to publish the peer review history of their article (what does this mean?). If published, this will include your full peer review and any attached files.

Reviewer #1: **Yes: **John M. Beggs

Reviewer #2: No

---

## [Editor Report · Acceptance letter]

15 Jan 2024

PCOMPBIOL-D-23-00845R1 

Revealing and reshaping attractor dynamics in large networks of cortical neurons

Dear Dr Beer,

I am pleased to inform you that your manuscript has been formally accepted for publication in PLOS Computational Biology. Your manuscript is now with our production department and you will be notified of the publication date in due course.

With kind regards,

Judit Kozma
